# Simultaneous Analysis of L-Carnitine and Acetyl-L-Carnitine in Food Samples by Hydrophilic Interaction Nano-Liquid Chromatography

**DOI:** 10.3390/mps8060145

**Published:** 2025-12-01

**Authors:** Cemil Aydoğan, Muhammed Ercan, Ziad El Rassi

**Affiliations:** 1Food Analysis and Research Laboratory, Bingöl University, Bingöl 12000, Türkiye; 2Department of Food Engineering, Bingöl University, Bingöl 12000, Türkiye; 3Department of Chemistry, Bingöl University, Bingöl 12000, Türkiye; 4Graduate School of Natural and Applied Sciences, Bingöl University, Bingöl 12000, Türkiye; mercan@bingol.edu.tr; 5Department of Chemistry, Oklahoma State University, Stillwater, OK 74078-3071, USA

**Keywords:** carnitine, HPLC, monolith, milk, nano-LC, dietary food supplements

## Abstract

L-Carnitine (L-CAR) and acetyl-L-carnitine (Acetyl L-CAR) are the essential cofactor compounds in lipid metabolism and are used in the treatment of various diseases. The European Food Safety Authority (EFSA) has reported that Acetyl-L-CAR contributes to normal cognitive function and has a beneficial physiological effect. Therefore, the sensitive separation and determination of L-CAR and Acetyl-L-CAR in foodstuffs can provide critical information. A notable trend in modern food analysis is the increasing use of miniaturized analytical columns with a narrow inner diameter (ID). In this study, a new, green analytical method for food analysis was developed to analyze L-CAR and Acetyl-L-CAR in food samples by nano-LC/UV with a hydrophilic monolithic 100 µm ID capillary. This is the first time that the preparation and application of a hydrophilic monolithic nano-column for the analysis of L-CAR and Acetyl-L-CAR in food samples by nano LC/UV has been reported. The hydrophilic monolith was prepared using in situ co-polymerization of glyceryl methacrylate (GMM) and ethylene dimethacrylate (EDMA). Following preparation and characterization, the hydrophilic monolith was used to analyze L-CAR and Acetyl-L-CAR in food samples, including three infant powdered milk samples and five supplements using nano LC/UV. The developed method was validated in terms of precision, sensitivity, linearity, recovery, and repeatability. The LOD and LOQ values were found to be in the range of 0.04–0.09 µg/kg, respectively. In short, the proposed method proved to be suitable for the routine analysis of L-CAR and Acetyl-L-CAR in food samples.

## 1. Introduction

L-Carnitine (L-CAR), as a zwitterionic molecule, is an essential nutrient which plays crucial role in human metabolism. Acetyl-L-carnitine (Acetyl-L-CAR) is an effective dietary supplement for dieting, which breaks down into L-CAR in the blood by plasma esterases. The body uses Acetyl L-CAR to transport fatty acids into the mitochondria for breakdown and energy production. The European Food Safety Authority (EFSA) reported that Acetyl-L-CAR contributes to normal cognitive function, having a beneficial physiological effect [1]. EFSA also reported the contribution of L-CAR to normal lipid metabolism [2]. Acetyl-L-CAR plays a key role in transporting fatty acids into mitochondria for energy production, which is vital for rapidly growing infants, especially preterm ones. It was used in supplements often adhering to stringent quality benchmarks, ensuring consistency and safety [3]. Both L-CAR and Acetyl-L-CAR are used as food supplements in foods for nutritional purposes, including functional foods and sports nutrition products. While L-CAR is not considered essential for children and adults, it is a vital nutrient for babies and should be included in infant formula [4]. However, the detection of L-CAR in infant powdered milk is difficult since it contains significantly low amounts of L-CAR. Many analytical methods have been developed and used for the analysis of L-CAR and Acetyl-L-CAR in food [5]. Conventional HPLC techniques are not suitable for analyzing food samples due to the low levels of L-CAR present as well as these techniques are neither green nor sensitive and are also time consuming. In this sense, more advanced sensitive and environmentally friendly analytical techniques are needed to analyze L-CAR and Acetyl-L-CAR in food samples.

In recent years, advanced chromatographic techniques have become increasingly prevalent in modern analytical chemistry. Among these techniques, nano-liquid chromatography (nano-LC), a miniaturized technique with significant potential for advanced food analysis [6]. This system is an improved version of the classical HPLC system and offers significant advantages over the classical HPLC system, such as low sample requirement, short analysis time, use of small amounts of reagents, compatibility with MS, high sensitivity, and rapid analysis. To ensure the greenness and sustainability of the method, it is of great significance to evaluate the environmental friendliness of analytical techniques. According to the AGREE assessment metrics, the most common quantitative greenness assessment tool with 12 green analytical chemistry (GAC) principles, AGREE score of nano-LC is 0.68 or higher, which means good greenness [7]. Nano-LC also offers many advantages over conventional analytical systems for food analysis, such as fast separation and high efficiency.

The miniaturization of analytical columns has gained significant momentum in liquid chromatography. Packed columns are widely used in nano-LC systems. However, preparing packed columns with low internal diameters (≤75 µm) is quite difficult due to the frit used and how the particles are placed in the column. Furthermore, their use is not long-lasting. Separation in packed columns is also based on diffusion, which negatively affects mass transfer. Monolithic columns have been developed as an alternative to packed columns and are widely used in food analysis [8,9]. Advantages of monolithic columns include easier preparation compared to particle-packed columns, no requirement for frit, and compatibility with low internal diameter columns are important advantages of monolithic columns. There is great interest in the use of advanced monolithic nano-columns, which are a cutting-edge technology in modern chromatographic science. Monolithic nano-columns have a large surface area and promise great potential for highly sensitive and green food analysis [6].

This study presents a new, environmentally friendly method for simultaneously analyzing L-CAR and Acetyl-L-CAR in food samples. A hydrophilic monolithic nano-column was prepared and used to develop the nano-LC/UV method. The analysis of L-CAR and Acetyl-L-CAR in three infant powdered milk samples and five different food supplement samples was then carried out using the developed hydrophilic monolithic nano-column.

## 2. Materials and Methods

### 2.1. Materials and Chemicals

L-Carnitine (>98%, Lot #SLBP4786V) and acetyl-L-carnitine (>96%) were purchased from Sigma-Aldrich Chemical (Milwaukee, WI, USA), as were cyclohexanol and dodecanol. Glyceryl methacrylate (GMM), ethylenedimethacrylate (EDMA), methanol (MeOH) and acetonitrile (ACN, 99% LC-grade) were purchased from Merck AG (Darmstadt, Germany). A nanoViper capillary (6041.5775) from Thermo Fisher Scientific (Sunnyvale, CA, USA) was used for the mobile phase delivery from pump to valve, which was purchased from BGB Analytik, Istanbul, Türkiye. A fused silica capillary with a 100 µm ID and a 363 µm OD (TSP-050375), a reducing union with 1/16′′ to 360 µm, SS, were also purchased from BGB Analytik, Istanbul, Türkiye. Two Quick Mount internal unions 360 µm (JR-C360QUPK4) from ViciValco (Brockville, ON, Canada) were used for a hydrophobic trap column connection with the analytical monolithic column. Pre-column extraction was performed using a trap column (PEPMAP 100 C18, 5 µm). The nano-column was connected to the detector’s flow cell with a 3 nm pore size using a Quick Mount Union (VICI Valco), 360 µm, 2–56 bore, 50 µm.

### 2.2. Instrumentation

The ProFlow nano-LC (CS-3500RS Nano ProFlow, 5041.0010A, Thermo Fisher Scientific, Sunnyvale, CA, USA) was used for the LC-UV experiments. This system includes both an autosampler with a WPS-3000TPL RS and a quick VWD-3400RS detector with a 3 nL flow cell. Solvent preparation and degassing were performed using a 6 L ultrasonic bath (WUC-D10H). Ultrapure water (18 MΩ cm resistivity) required for use in the analysis was produced by the water purification system from Millipore Corporation (Billerica, MA, USA) water purification system. Scanning electron microscopy (SEM, Zeiss (Oberkochen, Germany) Evo-50) was used to examine the surface morphology of the columns. Sample centrifugation was performed using a HERMLE Labortechnik GmbH centrifuge (Model Z 327K, made in Wehingen, Germany).

### 2.3. Silanization of Fused Silica Capillary

Silanization is a preliminary process that takes place before the polymerisation stage of the monolithic capillary column. This procedure was carried out as described in the published article [10]. In brief, the empty capillary column was first treated with 0.5 M NaOH for 15 min, after which it was washed with deionized water for a further 15 min. Next, methanol and a 50:50 mixture of TMSPM and methanol were passed through the column for 15 min, respectively. Finally, both ends of the column were covered with a septum and kept in a water bath at 35 °C for 18–20 h. The column was then removed from the water bath and washed with methanol for 15 min. This step aims to remove the structures that cannot attach to the inner wall of the capillary column. Finally, the column was treated with acetone for 15 min and then placed in an oven at 50 °C for 24 h to dry it completely.

### 2.4. Monolithic Column Preparation

A silanized fused silica capillary with an inner diameter of 100 µm and a length of 13 cm was used for the synthesis of a monolithic nano-column. To prepare the polymeric solution, the following were mixed in a 1.0 mL Eppendorf: 11.13% GMM (*wt*/*wt*), 12.11% EDMA (*wt*/*wt*), 38.01% cyclohexanol (*wt*/*wt*), 38.01% 1-dodecanol (*wt*/*wt*) and 0.70% AIBN (*wt*/*wt*). After being injected into the silanized capillary column, the filled capillary with the polymeric mixture was placed in a water bath at 75 °C for 4 h. The final hydrophilic capillary monolith was washed with an ethanol/water solution (80:20 *v*/*v*) for 2 h prior to use.

### 2.5. Sample Preparation

The sample preparation method was modified from a previously reported sample pretreatment method [11,12]. Three infant powdered milk samples (S1–S3) were purchased from local supermarkets while five different food supplement samples S4–S5 for Acetyl L-CAR and S6–S8 for L-CAR were purchased from health food shop in Bingöl (Türkiye). 250 mg of each solid sample or 250 μL of each liquid sample was placed into a falcon tube and diluted with 100 μL of acetic acid. At this stage, 2 mL of ultrapure water was added to the sample. In the next step, it was kept in a water bath at 35 °C for 5 min. At the end of this period, the supernatant formed in the falcon tube was removed, 2 mL of pure water was added, and the sample was centrifuged for 30 min at 3000× *g* to achieve complete separation at 4 °C. The supernatant in the falcon tubes was filtered through a 0.20 µm pore diameter filter using a syringe. The filtered sample was diluted by 1/2 and 1/4. The samples obtained in the last step were stored at 4 °C for analysis. The solutions were stirred for 10 min and then centrifuged. The supernatant was filtered and used in nano-LC analysis.

### 2.6. Method Validation

The performance of the method with the developed hydrophilic monolith was evaluated using LOD, LOQ and linearity. The mobile phase was composed of ACN and water. Standard solutions of L-CAR and Acetyl-L-CAR were prepared at concentrations of 1000 µg/mL in ACN/H_2_O and stored at 4 °C in a refrigerator. Calibration standards of L-CAR and Acetyl-L-CAR were prepared at different concentrations, including 0.01, 0,1, 1, 10, 100, and 1000 µg/mL using stock solutions. These standards were plotted against the respective internal standard (x) versus the peak area ratio (y). The LOD (*n* = 3) and LOQ (*n* = 5) were calculated using the spiked samples [13]. Spiked samples at three concentration levels were used for both recovery and repeatability studies. Spiking experiments and blank matrix tests for potential interferences and confirming the method’s selectivity, were performed according to published literature [14,15]. Matrix effect was performed using matrix-matched calibration curves, which are calibration plots created using standards prepared in the same matrix as the sample and comparisons with standard solutions. The concentration levels used for the spiking experiments covered a range representative of the expected levels of the analyte in real samples. The number of replicates performed at each level ensures statistical robustness. These steps may help ensure the robustness of the method when applied to diverse food samples.

## 3. Results and Discussion

### 3.1. Preparation and Characterization of the Hydrophilic Monolith with 100 µm ID

Monolithic columns have attracted significant interest in the field of food science and technology for use in food analysis. In a previous study, a graphene oxide-based monolithic nano-column with an ID of 100 µm was prepared and used to determine the geographic origin of 52 honey samples using micellar nano-LC and a new algorithm [16]. Considering the hydrophilic structure of L-CAR and Acetyl-L-CAR, this study involved preparing and using a hydrophilic monolithic nano-column with a 100 µm ID for the simultaneous analysis of L-CAR and Acetyl-L-CAR in food samples by nano-LC/UV. To prepare the hydrophilic monolith, GMM was used as the monomer because GMM provides a hydrophilic diol surface, which is promising for the hydrophilic interaction of the compounds. Ethylene dimethacrylate (EDMA) was used as the crosslinker, which may impart some non-polar characteristics to the surface. The polymerization mixture as detailed in Section 2 (see Section 2.4) was prepared and injected into the silanized fused silica capillary column. After preparation, the column stability was evaluated by measuring the column back pressure (Δ*P*_column_) as a function of flow rate, using ACN/H_2_O 80:20 (*v*/*v*) as the mobile phase. Appendix A shows the linear relationship (R^2^ > 0.9995) between the resulting back pressure (Δ*P*_column_) and nanoliter flow rate. This demonstrates the stability of the monolithic nano-column when using the mobile phase ACN/H_2_O 85:15 (*v*/*v*). Further details on the column morphology and hydrodynamic properties are provided in Appendix A. As can be seen, the column morphology was very homogeneous after washing, which proves the stability of the column. SEM images of the hydrophilic monolith are shown in Figure 1. The hydrophilic stationary phase was strictly coated on the inner surface of the column (Figure 1A), and both nano globules measuring less than 200 nm and pores measuring 3 µm could be obtained (Figure 1B). These results demonstrate that the prepared hydrophilic monolith with a 100 µm ID is distributed equally.

The hydrophilicity and non-polar properties of the monolith surface were investigated by examining the retention of two test solutes (thiourea and toluene) in a mobile phase containing a wide range of ACN. Figure 2 shows that the retention time of thiourea increases with increasing ACN content in the mobile phase, while the retention of toluene decreases. In terms of hydrophilic interactions, thiourea exhibits good retention at a high ACN content (e.g., 90%) in the mobile phase, indicating a strong hydrophilic interaction with the monolith surface.

The FT-IR spectra confirmed successfully the structure of the monolithic column. Appendix A shows the FT-IR spectra of the column. It can be seen that several enhanced absorption bands at 2923.14 cm^−1^, and at 2853.61 cm^−1^ were obtained and furthermore, the intense band of 1721.09 cm^−1^ is due to the stretching vibrations of the carbonyl group while the broad band in the range of 3349.90 cm^−1^ resembles the stretching of -OH groups. The measurement of the pore size and specific surface area of the monolithic columns were performed using nitrogen physisorption while Brunauer–Emmett–Teller method was used to calculate the specific surface areas of the columns. The surface area of the column was calculated as 141.2 m^2^/g. The repeatability and reproducibility of the developed column were performed by measuring the relative standard deviations (RSDs) of the retention factor of propylbenzene as the test compound (thiourea as void marker) using a mobile phase at 60% ACN. The RSDs of run-to-run, column-to-column and batch-to-batch test were less than 18%, 2.3%, 2.9% (*n* = 5), respectively.

### 3.2. Optimization of Chromatographic Conditions in Nano LC-UV

The chemical structures of L-CAR and acetyl-L-CAR are shown in Figure 3. Both have a carboxyl group and a quaternary ammonium group, which are hydrophilic groups that allow preferential partitioning in the water layer adsorbed at the monolithic surface using a hydroorganic mobile phase with a high ACN content. These compounds are difficult to retain on a reversed-phase column. Several hydrophilic columns were used to analyze these compounds. For example, a 4.6 mm ID hydrophilic column was used to separate L-CAR and Acetyl-L-CAR in milk and dairy products [17] while Kivilompolo et al. used a 2.1 mm ID hydrophilic column to analyze L-CAR and Acetyl-L-CAR derivatives [18]. In these studies, ACN and MeOH were mostly used as organic solvents for the compounds’ hydrophilic interaction. Hydrophilic interaction chromatography shows promise for separating L-CAR and Acetyl-L-CAR in samples. Considering the preliminary results regarding the solubility of the analytes, methanol: water was not suitable, whereas acetonitrile: water was found to be a suitable mobile phase and was selected for further studies. This mobile phase was used at a concentration of 60–95% (*v*/*v*) ACN/Phosphate-buffered solution (pH 3.4). Trifluoroacetic acid (TFA) at 1% (*v*/*v*) was added to the mobile phase as a counterion to provide better interaction between the hydrophilic monolith and the analytes. The retention times of L-CAR and Acetyl-L-CAR were investigated by varying the ACN content of the mobile phase. The presence of GMM as a hydrophilic monomer in the column structure allows hydrophilic interactions. As anticipated, the monolith interacts strongly with both L-CAR and Acetyl-L-CAR, retaining them using a hydroorganic mobile phase. Various flow rates ranging from 300 to 800 nL/min were applied to separate both L-CAR and Acetyl-L-CAR, with each standard injected individually. The retention time increased as the ACN content in the mobile phase increased from 65–95% (*v*/*v*), and the best separation was achieved using a mobile phase containing 92% (*v*/*v*) ACN. The best results were obtained using a mobile phase containing 92:8 (*v*/*v*) ACN/10 mM Phosphate-buffered solution (pH 3.4) at a flow rate of 400 nL/min. Figure 4 shows the chromatograms of the separation of both L-CAR and Acetyl-L-CAR using nano-LC/UV at a wavelength of 220 nm, which showed the maximum absorbance value. Conversely, the selected wavelength produced a good response, and this result is consistent with that reported in the literature [19]. Kinetic curves were obtained in the presence of different concentrations of the compound, allowing for promising retention of both L-CAR and Acetyl-L-CAR on the hydrophilic monolith. The results show that the monolithic column had a highly hydrophilic surface, indicating promising hydrophilic interactions of the compounds. This optimized method was then used to separate and analyze both analytes in samples using nano-LC/UV.

### 3.3. Loading Capacity

Sample loading was performed by injecting various concentrations of L-CAR and Acetyl-L-CAR, ranging from 0.01 to 5 mg/mL. The sample volume injected was 20 nL. The loading capacity of the developed hydrophilic column was calculated according to the corresponding peak width at half-height (W_1/2_) [20]. An increase of 10% was observed in the peak width at a small sample volume. Figure 5 shows the loading capacity test results for a 13 cm hydrophilic monolith. As can be seen, the W_1/2_ for L-CAR at 0.1 µg/mL increased by 17% over the peak width, while the W_1/2_ for Acetyl-L-CAR at 0.3 µg/mL increased by 13% over the peak width. The loading capacities for L-CAR and Acetyl-L-CAR were 0.55 mg/mL and 0.84 mg/mL, respectively. This indicates that the hydrophilic monolith under investigation has a relatively high loading capacity for both L-CAR and Acetyl-L-CAR.

### 3.4. Method Validation

The repeatability and reproducibility of the developed hydrophilic monolith were evaluated using RSD values obtained from the retention of thiourea as the test compound (toluene as the void marker) in the mobile phase containing 92% ACN. The RSD values were found to be in the range 0.9–1.2 for the same day and 0.7–1.7 for different days. The hydrophilic monolith exhibited promising repeatability and reproducibility. Table 1 shows the RSD values for both L-CAR and Acetyl-L-CAR, calculated using the peak areas of the compounds. The results demonstrate the good repeatability of the developed hydrophilic nano-LC/UV method. Three infant powdered milk samples and five different food supplements were studied, and recovery and repeatability were calculated by analyzing these spiked samples at three concentration levels. Repeatability, recovery, linear range and sensitivity were calculated using the samples in hydrophilic nano-LC/UV. Table 2 shows the calibration curves calculated using L-CAR and Acetyl-L-CAR standards. As Appendix A shows, the calibration data for L-CAR and Acetyl-L-CAR were more than 0.999 for both compounds. The LOD and LOQ values were found to be in the range of 0.04–0.09 µg/kg, respectively. These results revealed good linearity (R^2^ > 0.999) and sensitivity. Intra-day and inter-day precisions of L-CAR and Acetyl-L-CAR were found in the ranges of 1.9–2.4 and 2.7–3.1, respectively. No matrix effects occurred, or they were compensated completely. Sanchez-Hernandez et al. reported the determination of L-CAR in dietary supplements using capillary electrophoresis-tandem mass spectrometry, achieving an LOD value of 10 ng/mL for L-CAR [21]. In the current study, the LOD value was found to be 0.04 µg/kg, a more sensitive result than that found in the aforementioned study. Another study relating to milk-based infant formulas was performed using a Click Xlon hydrophilic UPLC column to analyze L-carnitine derivatives in milk-based infant formulas and healthcare products, in which the LOD value was found to be 25 mg/g for L-CAR [22]. As expected, the current method which uses a 100 µm ID hydrophilic monolithic column, allowed a better LOD level. Spiking blank samples with five replicates per level (*n* = 5) was used to determine the precision and accuracy of nano-LC/UV. Accuracy was found to be in the range of 88–105%, indicating that the developed method with a hydrophilic monolith demonstrated good performance in the sensitive separation and analysis of both L-CAR and Acetyl-L-CAR in the samples.

### 3.5. Analysis of L-CAR and Acetyl L-CAR in Food Samples

Various food samples, including infant powdered milk and different food supplement samples, were examined for L-CAR derivatives, including Acetyl-L-CAR [5,12], and the methods involved were based on conventional liquid or gas chromatography techniques. In the current study, a new, sensitive method was developed to analyze L-CAR and Acetyl L-CAR in three infant powdered milk samples (S1–S3) and three food supplements (S6–S8), and two food supplements (S4–S5). The current method uses nano-LC/UV with a narrow bore hydrophilic monolithic column of 100 µm ID, which was then applied to the samples to evaluate its suitability for determining L-CAR and Acetyl L-CAR in routine analysis. Table 3 shows the L-CAR and Acetyl L-CAR content in the samples. As can be seen, no Acetyl-L-CAR was found in sample S2. On the other hand, a high L-CAR content was found in sample S1, at around 25.02 mg/kg, while the other samples showed 0.48 and 7.7 mg/kg. These results demonstrate that the developed nano-LC/UV method is sensitive and promising for food analysis and could also be applied to other food samples to determine L-CAR and Acetyl-L-CAR.

### 3.6. Greenness Assessment

Several metric tools have been proposed that allow methods to be evaluated according to the principles of green analytical chemistry using specific criteria. The most widely accepted of these metrics is AGREE [23], which was used to obtain a final score of 0.79 out of 1.00 using the nano-LC analytical methodology (Appendix A). This methodology was developed in line with the principles of green analytical chemistry. The results showed that the developed method with Nano-LC is an excellent assay for developing new, sustainable analytical methodologies.

## 4. Conclusions

The development of a new method for the analysis of L-CAR and Acetyl L-CAR by nano LC/UV was studied by altering the composition of the mobile phase, particularly the content of the aqueous and organic phases under hydrophilic conditions. The influence of these changes on the retention factors and the separation process was then evaluated. The optimization of the acetonitrile content of the mobile phase markedly improved the separation and resulted in faster analysis. Overall, the new method drastically reduced solvent consumption and waste production, with a reduction factor of up to 2000 compared to classical HPLC. In conclusion, a new green and sensitive method was developed for the simultaneous analysis of L-CAR and Acetyl L-CAR using nano LC/UV with a hydrophilic monolithic stationary phase. The proposed method was successfully validated in terms of sensitivity, linear range, repeatability and recovery, and improved sensitivity was achieved for the compounds compared to other previously described studies. The developed strategy, which uses a hydrophilic monolithic capillary, has many advantages over existing methods. The development and utilization of such green, sensitive analytical method is necessary for advanced food analysis.

## Figures and Tables

**Figure 1 mps-08-00145-f001:**
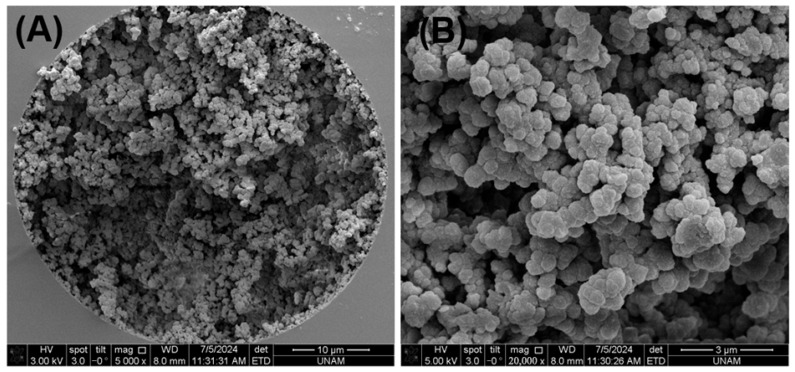
SEM images of the hydrophilic monolith with 5000 magnifications in (**A**) and 20,000 magnifications in (**B**).

**Figure 2 mps-08-00145-f002:**
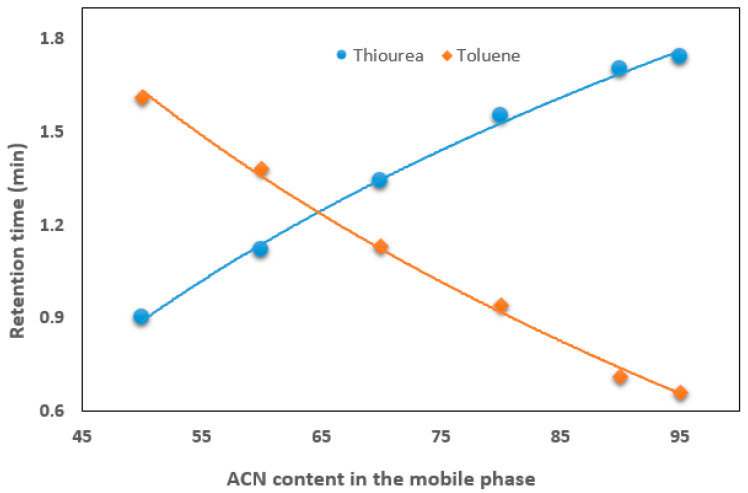
Effect of ACN content on the retention time of thiourea and toluene.

**Figure 3 mps-08-00145-f003:**
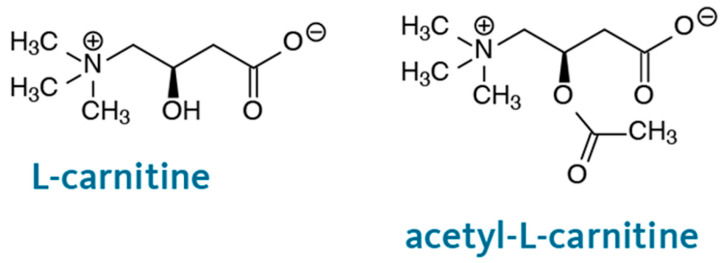
Chemical structure of L-CAR and Acetyl-L-CAR.

**Figure 4 mps-08-00145-f004:**
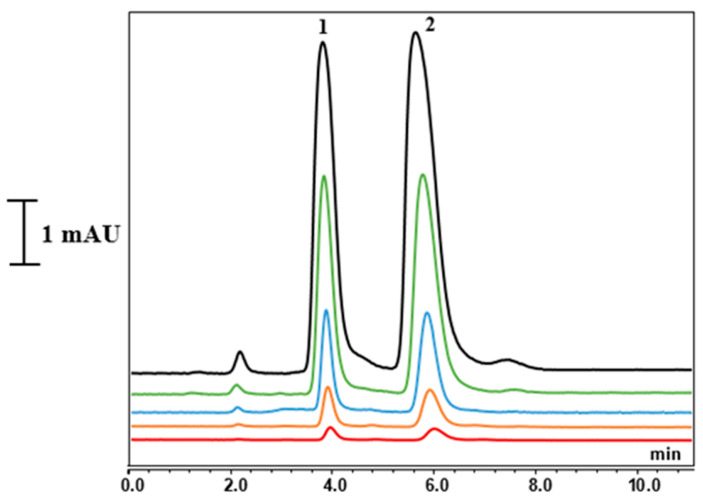
Separation chromatograms of L-CAR and Acetyl-L-CAR with different concentration. Chromatographic conditions: mobile phase: ACN/Water with phosphate-buffered solution (92:8, *v*/*v*) with 0.1% TFA. Flow rate: 400 nL/min. 220 nm: Peaks: (1) L-CAR, (2) Acetyl-L-CAR.

**Figure 5 mps-08-00145-f005:**
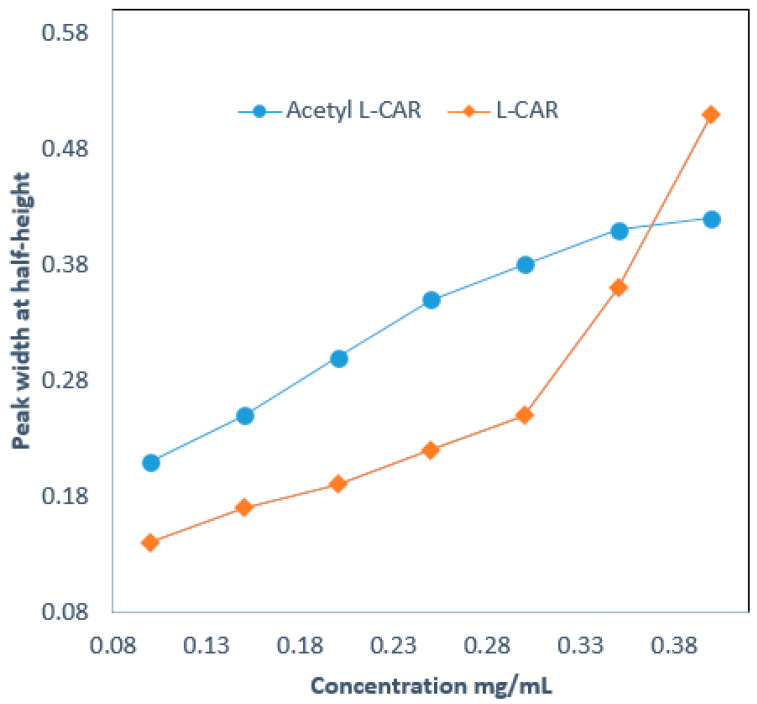
Loadability tests of a 13 cm hydrophilic monolith using L-CAR and Acetyl-L-CAR.

**Table 1 mps-08-00145-t001:** Results of the repeatability study of nano-LC-UV method.

Peak	Analyte	Retention Time	Peak Area	Retention Time	Peak Area
1	L-CAR	3.9	10.2	4.2	10.4
2	Acetyl-L-CAR	6.1	12.7	6.3	13.1

**Table 2 mps-08-00145-t002:** LODs, LOQs, Recoveries and Correlation coefficients (R^2^), for L-CAR and Acetyl L-CAR.

Compound	Recovery (*n* = 5)						Recovery (*n* = 5)					
	0.05 µg kg^−1^	0.5 µg kg^−1^	5.0 µg kg^−1^	LOD	LOQ	R^2^	0.05 µg kg^−1^	0.5 µg kg^−1^	5.0 µg kg^−1^	LOD	LOQ	R^2^
**L-CAR**	97	89	98	0.06	0.08	0.9982	96	104	98	0.06	0.08	0.9998
**Acetyl L-CAR**	93	95	94	0.04	0.09	0.9985	89	96	95	0.08	0.09	0.9996

LOD: Limit of detection. LOQ: Limit of quantification.

**Table 3 mps-08-00145-t003:** Contents of L-CAR and Acetyl L-CAR in food samples.

Samples	S1	S2	S3	S4	S5	S6	S7	S8
**L-CAR**	25.02 mg/kg	0.48 mg/kg	7.7 mg/kg	-	-	870 mg/cap	441 mg/cap	122 mg/cap
**Acetyl L-CAR**	14.4 mg/kg	ND mg/kg	0.19 mg/kg	1090 mg/cap	488 mg/cap	-	-	-

## Data Availability

The original contributions presented in the study are included in the article, and further inquiries can be directed to the corresponding author.

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
