# Peer review of "Simultaneous Analysis of L-Carnitine and Acetyl-L-Carnitine in Food Samples by Hydrophilic Interaction Nano-Liquid Chromatography"

_mps, 2025, doi:10.3390/mps8060145_

Round 1
Reviewer 1 Report
Comments and Suggestions for Authors
This study developed a novel method based on hydrophilic interaction nano-liquid chromatography (nano-LC/UV) for the simultaneous analysis of L-carnitine and acetyl-L-carnitine in food samples. Using a hydrophilic monolithic column with 100 μm internal diameter, the method demonstrates significant advantages in greenness and sensitivity, and represents the first report of such a column for simultaneous analysis of these compounds in food matrices. The study design is reasonable, method validation parameters (linearity, sensitivity, precision, recovery) meet acceptable criteria, and successful application to real samples (infant formula and dietary supplements) is shown. However, the manuscript has shortcomings in evaluation of matrix effects, sample representativeness, and method robustness validation. Major revision is recommended. The authors should supplement relevant experimental data, deepen the discussion, and clarify some methodological details.
- The significance of the study is not sufficiently articulated. While EFSA's recognition of acetyl-L-carnitine is mentioned, the practical application prospects in food safety, infant nutrition assessment, or quality control of dietary supplements are not thoroughly discussed. Please expand on this.
- The novelty of the method should be highlighted further, emphasizing the technical advantages of the 100 μm ID monolithic column in nano-LC and providing quantitative comparisons with existing methods (e.g., HPLC, CE), particularly in terms of green chemistry metrics.
- The characterization of the monolithic column is incomplete. Only SEM images and backpressure tests are provided. It is recommended to supplement with pore size distribution, specific surface area data, FT_IR and column efficiency (theoretical plate number) measurements.
- The optimization process of chromatographic conditions is vaguely described. Details on the optimization of mobile phase composition (ACN ratio, buffer concentration, pH), flow rate, etc., should be provided, along with experimental rationale.
- Method selectivity is not sufficiently verified. The potential interference from other compounds (e.g., amino acids, organic acids) in complex food matrices was not investigated. Spiking experiments or blank matrix tests are recommended.
- Matrix effect was not evaluated. This is a major omission. Given the complexity of food matrices (especially powdered milk), significant matrix effects are likely. Assessment via comparison of standard solution and matrix-matched calibration curves is necessary.
- The sample preparation procedure is overly simplified. Only "centrifugation and filtration" are mentioned, lacking details (e.g., centrifugal speed/time, filter type/pore size). A detailed description is essential for method reproducibility.
- The sample representativeness is insufficient. Only three infant formula and five supplement samples from single sources (local supermarkets and shops) were analyzed. Increasing sample number and covering diverse brands/origins is advised to demonstrate method universality.
- The recovery study design needs improvement. The spiking concentration levels, number of replicates per level, and whether matrix-matched calibration was used for recovery calculation should be clearly stated.
- The LOD/LOQ calculation method is non-standard. The manuscript mentions "calculated using spiked samples" but does not specify the formula or criteria (e.g., signal-to-noise ratio). Adherence to IUPAC guidelines and clarification of the calculation method are required.
- Repeatability and reproducibility data are not presented. Only RSD ranges are mentioned without specific data tables or chromatograms. Presenting intra-day and inter-day precision results in a table is recommended.
- The quality of Figures 4 and 5 is poor. The chromatograms have low resolution, making it difficult to discern peak shapes and resolution. Higher-resolution images or original data plots are suggested.
- The discussion of real sample analysis results is superficial. Only content data are reported without discussion on reasons for variations among samples or comparison with literature values. Deeper analysis is needed.
- The greenness assessment lacks supporting data. Although solvent consumption of nano-LC is mentioned, complete green chemistry metrics (e.g., E-factor, AGREE score) calculations or comparisons are not provided.
- The conclusion section should be expanded to summarize the core advantages and limitations of the method, and suggest future improvements or application extensions.
- All the figures and tables need to be standardized and beautified.
The English could be improved to more clearly express the research
Reviewer 2 Report
Comments and Suggestions for Authors
The subject undertaken by the Authors is impotrant from the points of view of development of food analysis methodology as well as employnig more environmental friendly approach to instrumental analysis by using miniaturized liquid chromatography. The rationale of the research is clearly described in the Introduction section. The experiments conducted and the results obtained are more or less clearly presented, although some sections should be more clearly presented and some details needs correction:
- In the section 2.1 Materials and Chemicals :
-quick instead of quic
- detector flow cell should be rather of 3 nL of volume, not of 3 nm pore size
2. Section 2.4 - was the polymerization mixture sparged with an inert gas like helium or nitrogen?
3. Section 2.5 - please describe the sample preparation methodology with more details and ensure that the masses and volumes are given properly (100 µL of acetic acid added to the 250 mg of sample and stirred in a beaker looks a bit strange).
4. Section 2.6 - what was the matrix for the spiked samples?
5. Section 2.6 - I could not find a link to the Supporting material containing Figure S1.
6. Section 3.2 - as for mobile phase composition: the Authors mention that it was ACN/water WITH ADDITION of 10 mM of phosphate buffer solution (pH 3.4) or it should be that a water part of the mobile phase WAS the 10 mM phosphate buffer solution? Next the Authors mention about addition of 1% of TFA, then that a final mobile phase was 92/8 ACN water with phosphate buffer solution. Please correct this section.
7. Section 3.3 - please describe in more details the procedure of loading capacity calculations. Please note, that in one sentence the Authors write about "peak width at half-height (W1/2)" and "peak width" while it should always be "peak width at half-height (W1/2)". Please correct this section to provide more precise discussion.
Round 2
Reviewer 1 Report
Comments and Suggestions for Authors
Although the authors made efforts to revise the article, the results were not satisfactory. Figures 1 and 5, as well as Tables 1-3, must be modified to be more aesthetically pleasing. Figures 1 and 5 must be plotted using Origin.
Comments on the Quality of English LanguageThe English could be improved to more clearly express the research
Reviewer 2 Report
Comments and Suggestions for Authors
The manuscript was improved significantly. However, still some corrections need to be done:
- In the 3.2 section there is a partially corrected sentence: "This mobile phase was used at a concentration of 60–95% (v/v) ACN/Water with phosphate buffer solution, with the addition of a 10.0 mM phosphate buffer solution (pH 3.4)" Please correct it. I think the mobile phase could be simply described as: "ACN/10 mM phosphate buffer (pH 3.4)".
- As I did not have a chance to see Supplementary file during my first review, I did not see the calibration curve. I do not think it is properly prepared to validate the method. Please note that the second point represents solution 50x higher concentration than the first one, third point is for only 2x higher concentration than the second one, then the fourth one is for 100x higher concentration than the third one and finally last point is for 10x higher concentration than fourth one. It looks strange. For validation please prepare at least 6 points and the concentrations should be more or less evenly distributed across the chosen concentration range (0.01-1000 ug/mL), for example according to the scheme: 0.01-0.1-1-10-100-1000 ug/mL, of course if so broad range is really needed.
Round 3
Reviewer 2 Report
Comments and Suggestions for Authors
Very nice! I accept the manuscript in present form!
Congratulations!